# Effect of Fluoride Varnish in Preventing Dental Caries of First Permanent Molars: A 24-Month Cluster Randomized Controlled Trial

**DOI:** 10.3390/ijerph192416656

**Published:** 2022-12-11

**Authors:** Zhaoyou Wang, Wensheng Rong, Tao Xu

**Affiliations:** 1Department of Stomotology, Peking University Third Hospital, Beijing 100191, China; 2Department of Preventive Dentistry, Peking University School and Hospital of Stomatology, Beijing 100081, China

**Keywords:** fluoride varnish, first permanent molars, dental caries, randomized controlled trial

## Abstract

Background: Caries is a prevalent health problem. This study evaluated the effect of fluoride varnish in preventing dental caries of first permanent molars. Methods: The study was designed as a stratified cluster randomized controlled trial, with classes as the unit of randomization. Classes stratified by district were followed for 24 months. All eligible children of the selected classes were included for the trial. The children in the test group were biannually applied fluoride varnish. The outcomes were measured at an individual level. Results: In total, 107 classes (51 in the test group, 56 in the control group) were recruited for the trial. Of the 5397 participants, 5005 children (2385 in the test group, 2620 in the control group) completed the study. At the 24-month follow-up, the mean decayed and filled surface increment of the first permanent molars of the children in the test group was significantly lower than that of the children in the control group (0.38 versus 0.61). The caries incidence of the first permanent molars in the test group was 17.0%, while that of the control group was 23.7%, with a PF of 28.3%. Conclusions: Semi-annual application of fluoride varnish is effective in reducing the caries increments of first permanent molars.

## 1. Introduction

Over the last four to five decades, the prevalence and severity of dental caries of primary and permanent dentitions have declined in a number of countries [1,2,3]. Despite this achievement, the prevalence and severity of dental caries remain too high at a world level, and affect many children [1,2]. Dental caries has also presented a significant health challenge in China in recent decades [4]. From 2005 to 2015, data from the Chinese National Oral Health Survey showed that the prevalence of dental caries in the permanent dentition of 12-year-olds increased from 28.9% to 38.5%, and the mean decayed, missing and filled teeth (DMFT) index rose from 0.54 to 0.86 [4,5]. “The Healthy China Plan for 2030”, issued by the General Office of the State Council of China, stated that the objective was to control the prevalence of dental caries in 12-year-olds to under 30% by 2025, and to 25% by 2030 [6,7]. Since 2008, the Chinese government has allocated millions of Chinese yuan annually to the promotion of the oral health of school-aged children, which covers oral health education, oral hygiene instruction, and pit and fissure sealants for 7–9-year-old children [8]. However, Chinese local epidemiological surveys on caries reported that more than 90% of dental caries are confined to first permanent molars among school-aged children [9,10,11,12], and the prevalence of caries in first permanent molars had already reached approximately 20% among children aged 7–9 years [9,10,13]. Judging from these studies, first permanent molars are the most susceptible to caries, and approximately 20% of the teeth had already decayed before they reached the indication for sealants. The first permanent molar is particularly prone to dental caries within a short period after eruption in the mouth [14]. The vulnerability is mainly attributed to high plaque accumulation (as the complex fissures are partly covered by gingiva for a considerable period of time) [15,16], incomplete post eruptive maturation of the enamel [17] and lack of parental awareness of the tooth emergence [18]. Therefore, first permanent molars are crucial for caries prevention, and should be protected from the onset of tooth eruption. Pit and fissure sealants are able to prevent the dental caries of partially erupted molars [19,20,21,22]. Regrettably, the difficulty of humidity control for erupting teeth leads to a low sealant retention rate, and requires periodic replacement to maintain the sealants’ caries-preventive effect [19,20,23]. In addition, pit and fissure sealants are technique-sensitive and require sophisticated dental equipment and trained operators [24,25], which limits their popularization in remote and rural areas. The Cochrane database indicates that fluoride varnish is effective in reducing tooth decay in both primary and permanent dentitions [26]. Fluoride varnish is also considered to be a very safe dental product [27]. It does not have an unpleasant taste, nor does it pose a risk of fluoride over-ingestion in the form of acidulated phosphate fluoride (APF) gels [28]. In recent decades, some scholars have focused their attention on the effect of fluoride varnish in preventing dental caries of first permanent molars [29,30,31,32,33,34,35,36,37,38]. Most clinical trials have used positive controls [31,33,34,39]. However, these trials were unable to evaluate the actual effect of fluoride varnish. While some investigators used placebo or non-treatment controls, there is no consensus among the authors on the caries-preventive effect of fluoride varnish on first permanent molars [29,32,36,37,38]. To our knowledge, there have been very few studies on the evaluation of fluoride varnish on caries prevention in first permanent molars in China, especially on a large sample scale with a long follow-up. The aim of the present study was to evaluate the caries-preventive effect of a semi-annual application of fluoride varnish on first permanent molars. The null hypothesis was that fluoride varnish would be unable to prevent dental caries in first permanent molars.

## 2. Materials and Methods

### 2.1. Trial Design

This study employed a stratified cluster randomized controlled trial design, with classes as the unit of randomization. The classes were stratified by districts. The duration of the study was 24 months. The study adhered to the CONSORT and extension guidelines. The approval of the design and procedure of this trial was obtained from the Ethics Committee of the Chinese Stomatological Association (approval number: 2014-003), and retrospectively registered in the Chinese Clinical Trial Registry (ChiCTR-IIR-17013897. Registered 13 December 2017, http://www.chictr.org.cn/showproj.aspx?proj=23230 (accessed on 16 October 2022)). The trial was also approved by the local education bureaus and the school administrations.

### 2.2. Participants

The children were recruited from three low-fluoridated county-level cities (Dahua, Linxia and Linxiang) in China. Dahua is located in the southern part of China, Linxia in the northwest, and Linxiang in the middle. Individuals of Han ethnicity live in Linxiang, and many minority groups reside in Dahua and Linxia. They have quite different dietary constructions and living habits. The fluoride level in the local water supply of these three cities is under 0.2 mg/L. These county-level cities belong to rural areas, and have poor dental care due to the lack of dental manpower and sophisticated dental equipment. Public health measures, including pit and fissure sealants, were not commonly applied in these cities.

The children were recruited by three coordinators. The three coordinators were part of the research team, who were all experts in preventive dentistry in China. However, they did not take part in the dental caries examination, data recording or analysis. First, all districts of the county-level city were given computer-generated numbers, and those with even numbers were selected. Second, all grade one classes of large-scale public primary schools (≥450 students) of the selected districts were invited.

In an attempt to increase the level of participation and lower the rate of drop-out, the heads of the schools organized a parent meeting one month prior to the start of the study, to inform parents about the trial. Families were informed about the research, and consent forms were disseminated. Parents were also informed that their child’s participation was voluntary throughout the study, and that they could withdraw from the study at any time without any consequences. If the study participant could not follow the study protocol, he or she would be excluded from the study by the investigators. Any adverse events occurring during the study course were carefully recorded and followed.

The inclusion criteria required that the healthy child be 6–7 years old with no acute or chronic systematic disorders, no gingivitis or ulcers, and no asthma or allergic history to colophony (an ingredient in the fluoride varnish), and with no medication. The child could not take part in other trials during these 24 months. A signed parental consent form along with the child’s written assent was submitted. Children with hypoplastic defects, fluorosis or pit-and-fissure-sealed first permanent molars were excluded.

### 2.3. Sample Size

The minimum sample size needed was calculated to be 614 children per group, with 90% power to detect a difference between the group proportions at a two-sided alpha of 0.05. The caries prevalence in the test group was assumed to be 9%, and in the control group 15% [10,26]. Given an anticipated drop-out rate of 30%, the total sample size required was approximately 800 in each group. To further evaluate the effectiveness of fluoride varnish on preventing dental caries in three county-level cities separately, the study aimed to enroll approximately 4800 children in total.

### 2.4. Interventions

According to the answers of the questionnaire, oral health education had been given to all children and their parents each year in the classroom, including instructions on tooth brushing and dietary counselling. All children were encouraged to brush their teeth twice a day with fluoride toothpaste (containing 0.10% NaF) throughout the study. Oral hygiene instructions were repeated every 6 months, 4 times in total.

The children in the test group were scheduled for topical application of fluoride varnish at baseline, and then every 6 months, for a total of 4 applications during the 24-month study course. The fluoride varnish was applied by two dentists and assistants at schools in each county-level city. Duraphat^®^ (Colgate-Palmolive (UK) Limited, Waltrop, Germany) was used in this study, which is a fluoride varnish containing 5% sodium fluoride (NaF) (2.26% F-) in an alcohol suspension of natural resins [40]. It was commercially available in China in 2012. Every child was given a 0.25 mL fluoride varnish according to a standard card, corresponding to 5.65 mg F- per application. The teeth were isolated with cotton rolls and dried with swabs; the varnish was painted on all accessible tooth surfaces of the first permanent molars using a small disposable brush, and air-dried. The rest of the varnish was applied on other teeth of the oral cavity. The children were instructed not to drink or eat for at least two hours, and not to brush their teeth that day.

### 2.5. Randomization and Blinding

The randomization of participants was based on class. The class randomization was carried out by school administrators according to coin-flipping results, independent of the research team. Children from the same class were assigned to the same group. Each child was given an identification (ID) number at the first visit, and identified by this ID number throughout the study. 

Due to the physical nature of Duraphat, the scope of blinding was limited to the examiners and assistants for data recording and data analyst. The varnish providers and their assistants were informed of the allocation. However, they did not take part in the dental caries examination, data recording or analysis. The participants were very likely to be aware of the allocation due to the presence or absence of fluoride varnish. Allocation lists were revealed after study completion.

### 2.6. Data Collection

All parents were required to complete a questionnaire during the parent meeting at the first visit. The questionnaire included the child’s demographic characteristics (sex and birth date) and the child’s oral-health-related behavior (frequency of tooth brushing and sweets intake). The sweets include candies, sweet carbonated drinks or soft drinks, sweet biscuits or cakes, sweet milk or milk tea as well as any other sweet snacks.

The participants were examined at school, supinated, and under artificial light, by using plane surface dental mirrors and CPI probes. No radiography examinations were performed. After tooth brushing, the surfaces of the teeth were dried with cotton rolls and swabs. The eruption stages of first permanent molars were recorded. The caries status of the primary and erupted permanent teeth were recorded according to World Health Organization (WHO) criteria [41] at baseline and at the end of the 24 months. After the caries examination, a report was sent to the child’s caretakers to inform them whether the child needed treatment by the examining dentists.

The clinical dental examinations were undertaken by experienced dentists. Six examiners were used across the study, and were involved in all years of the 24-month study course. The examiners were trained and calibrated in advance of the baseline oral examination, and recalibrated before the final examination. A 5% random sample was re-examined during each clinical examination, to measure intra-examiner reproducibility. Compared with the reference examiner’s results, the six examiners demonstrated good reliability for caries diagnosis, with Cohen’s weighted kappa values >0.8 at baseline and at the 24-month follow-up examination. The intra-examiner consistencies reached over 90.0% agreement for all examiners.

### 2.7. Outcome Measures

Outcomes were measured at the individual level. The primary outcome measure for this study was the caries increment at the surface level (decayed and filled surfaces (DFS)) of the first permanent molars at the 24-month follow-up. The secondary outcome measure was the caries incidence of the first permanent molars.

### 2.8. Statistical Analysis

Only those children who completed the 24-month follow-up were included in the statistical analysis. The data were analyzed by SAS version 9.4 (SAS Institute Inc., Cary, NC, USA). Descriptive analysis was conducted to summarize the baseline characteristics of the study population. The outcomes between the two trial groups were compared by using linear regression for continuous variables and logistic regression for categorical variables over the 24-month course. A generalized linear model (GLM, SAS Proc Genmod) was employed to adjust for factors that may have influenced the outcome, based on previous studies (age, sex, baseline caries experience, frequency of tooth brushing, and frequency of sugar consumption) [30,42,43]. The statistical significance level for all tests was set at 0.05. The caries preventive fraction (PF) of the first permanent molars was derived as PF = (IC-IT)/IC (IC: caries incidence of the first permanent molars in the control group; IT: caries incidence of the first permanent molars in the test group). PF indicates the caries reduction rate due to the application of fluoride varnish.

## 3. Results

The trial was implemented from 20 October 2014 to 21 December 2016. Between 20 October and 10 December 2014, a total of 5583 children were assessed for eligibility (Figure 1) in 107 classes. Of these children, 186 were excluded mainly due to declined participation. Of those eligible children, 2657 were randomly allocated to the test group, and 2740 to the control group.

In total, 5005 children completed the entire course. The descriptive statistics at baseline for the children who completed the 24-month intervention are presented in Table 1. The groups were balanced with respect to demographic characteristics, oral hygiene, dietary habits and caries experience at baseline. The drop-out rate of the test group was 10.2% (272/2657), and the rate for the control group was 4.4% (120/2740).

At the 24-month follow-up examination, 98.5% of the first permanent teeth had fully erupted, with no group difference. After considering the confounding factors, the mean DFS increment of the first permanent molars of the children in the test group was significantly lower than that of the children in the control group (0.38 versus 0.61; *p* = 0.001, 95%CI: 0.11–0.44) (Table 2 and Table 3). The caries incidence of the first permanent molars in the test group was 17.0% (406/2385), while the incidence in the control group was 23.7% (622/2620) (Table 2).

The comparison of the incidence in first permanent molars between the children in the control group and in the test group yielded a significant odds ratio (OR) of 1.23 (95% CI: 1.03 to 1.48, *p* < 0.001), meaning that the odds of caries incidence in the children in the control group was 1.23 times higher than the test group (Table 4). Statistically significant results were found in the confounding factors concerning age, sex, baseline DFS and frequency of sugar consumption. The results suggested that: the older the participants, the higher the risk for caries; the females had a higher risk for caries; and the higher the sugar intake, the higher the risk for caries (Table 3 and Table 4). The PF was 28.3%. 

Only one child complained about the taste of the fluoride varnish. In that case, the taste did not cause any nausea or vomiting. There were other adverse effects (e.g., swelling, burning, itching, or soreness and rash) related to the application of fluoride varnish, as reported in the literature [27].

## 4. Discussion

The study evaluated the effect of fluoride varnish over a large group of children with diverse diets and living habits. The principle finding in this trial was that 5% NaF varnish significantly reduced the new caries increment of first permanent molars when applied semi-annually for 24 months. Therefore, the null hypothesis was rejected. This finding is in agreement with some clinical trials conducted both in China and overseas [29,32,37]. However, Hardman and Milsom did not find a significant caries reduction by semi-annual application of 5% NaF varnish in school-aged children. It has been suggested that the high drop-out rates and lower-than-expected caries increment may be the possible explanations [36,38]. Likewise, Ninosca reported that positive findings could only be observed when fluoride varnish was applied every three months as opposed to applying it every six months, after a one-year study course. A relatively short follow-up period may not ascertain the effect of a biannual application of fluoride varnish [30].

Our study found that 5% NaF varnish prevented 28.3% of the dental caries in first permanent molars over a course of 24 months. With a similar application frequency and study duration, Holm et al. reported that the caries PF was 56% among children aged 5 years and 9 months [37], which had a higher effectiveness than that found in our study. Holm’s study only assessed the caries-preventive effect of a topical application of fluoride varnish on the occlusal surfaces of newly erupted first permanent molars [37]. In addition, with an earlier application of fluoride varnish, a better caries-preventive outcome may be observed. A similar study was conducted by Zhang et al. among 200 children aged 6–8 years. It was reported that the caries incidence of the first permanent molars was 9.38% in the test group, and 27.88% in the control group [29]. The caries PF was 66.4% [29], which was significantly higher than in our study. This may be due to the difference in participants between their study and our study. In Zhang’s study, only children with four fully erupted first permanent molars were recruited [29], whereas in our study, children with partially erupted first permanent molars were also included. Furthermore, the sample size in their study was much smaller than that in our study, which may also explain the different results between the two studies. 

The prevalence of dental caries was relatively low at baseline in this study, compared with the results of some previous studies conducted among high-caries-risk populations [32,44]. In addition, the oral health education and oral hygiene instruction given to all the children may have had an effect on reducing the dental caries incidence in the study population. These may have in turn lead to a comparatively lower preventive fraction.

However, there were some limitations to the study. Firstly, the estimation of the sample size was not adjusted for the clustering effect of children within the same class. If taking into account clustering, the number should be multiplied by [1 + (m − 1) ×ICC]. The intraclass correlation (ICC) was estimated to be 0.2, and 40 children were expected in each class (m = 40) [45]. The adequate sample size should be 1424 children per group. Therefore, a sufficient number of children had already been recruited for the trial. Secondly, the intention-to-treat analysis was not employed for data analysis. The per-protocol analysis may enlarge the effect of fluoride varnish. Thirdly, a much higher drop-out rate was observed in the test group than in the control group (10.2% vs. 4.4%). The potential no-return of some participants can cause data fluctuation, which might have influenced the results of the present study, mainly concerning the PF value. Fourth, the participants were very likely to have known their allocation, which may have influenced the behavior of the study subjects in a manner that could have influenced the outcome. In addition, the children’s oral health-related behavior, as well as whether the children received any dental preventive interventions or treatment in dental clinics, were not followed during the 24-month study course. Therefore, it is not clear to what extent these affected treatment effectiveness. Lastly, the clinical trial was registered after the first participant was enrolled.

This clinical trial provides some support for the use of fluoride varnish for the prevention of dental caries among school-age children. Previous studies found that the semi-annual application of fluoride varnish had a similar caries-preventive effect compared to resin-based sealants on erupting first permanent molars [31,46]. From this perspective, fluoride varnish was shown to be clinically superior to pit and fissure sealants, to some extent. Fluoride varnish is safe, effective and easy to apply. Strict humidity control is not emphasized for the application of fluoride varnish, which is especially helpful for erupting teeth. Additionally, fluoride varnish does not require sophisticated dental equipment. Furthermore, children can easily cooperate with the application, due to the low application frequency and the short time needed for each application [28]. Therefore, the semi-annual application of fluoride varnish, in combination with oral health education and oral hygiene instruction, could be considered a public health plan for children aged 6 to 7 years. It can be offered in schools as a part of health promotion programs from the first grade, which will effectively control caries progression during the eruption process of first permanent molars. Once the caries increment of the first permanent molars is reduced among school-aged children, the caries increment of permanent dentition would be greatly reduced as well. Children in rural and remote areas have had more serious caries experience than those in cities [4,5]. However, the lack of dental manpower and sophisticated dental equipment makes conventional dental care either unavailable or unaffordable for these families. Therefore, topical application of fluoride varnish can be considered a desirable intervention for oral health promotion in these areas.

Dental caries and periodontal disease have historically been considered the most important global oral health burdens. Dental caries is still a major health problem in developing and developed countries, especially affecting school-age children [47]. Successful caries-preventive measures for school-age children thus have a substantial effect on burden, and dramatically diminish the costs associated with treatment. If 28.3% of dental caries were avoided, it would substantially reduce the need for dental treatment, and benefit children’s oral health conditions. Future efforts should include a cost-efficient analysis of the semi-annual application of fluoride varnish. Whether our results can be generalized to children in other county-level cities, different caries-risked populations or children in cities is uncertain. 

## 5. Conclusions

In conclusion, the results of this study suggest that the semi-annual application of 5% NaF varnish is effective in reducing caries increments during the eruption process of the first permanent molars.

## Figures and Tables

**Figure 1 ijerph-19-16656-f001:**
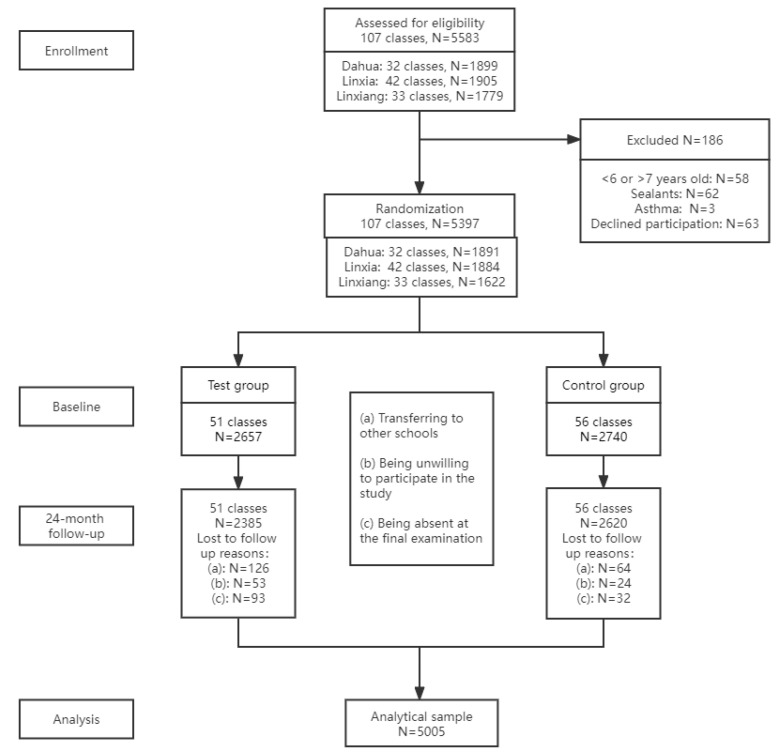
Flowchart of the participants of the trial.

**Table 1 ijerph-19-16656-t001:** Descriptive statistics at baseline for children who completed the intervention.

Variables	Test Group (*N* = 2385)	Control Group (*N* = 2620)
Age(years)		data
Mean(S.D.)	6.80(0.42)	6.85(0.42)
Sex(%)		
Male	55.2	53.3
Female	44.8	46.7
Frequency of tooth brushing(%)		
≥2/d	28.7	29.2
<2/d	71.3	70.8
Frequency of sweets intake(%)		
≥1/d	42.5	44.7
<1/d	57.5	55.3
Caries experience of primary dentition		
Prevalence(%)	87.3	85.7
Eruption stages of first permanent molars(%)		
0: Fully erupted occlusal surface and fully exposed crown, established antagonist contact	42.7	42.2
1: Fully erupted occlusal surface, partially exposed crown	13.4	13.2
2: Partially erupted occlusal surface	9.6	10.0
3: Only cusp erupted	2.0	2.0
4: No eruption	32.3	32.6
Caries experience of first permanent molars		
Mean DFS (S.D.)	0.02(0.20)	0.03(0.23)
Prevalence(%)	1.7	2.1

**Table 2 ijerph-19-16656-t002:** Descriptive analysis of the caries increment of the first permanent molars over a 24-month study course.

	Title 2	Title 3
Mean DFS (S.D.) ^1^	0.38(1.21)	0.61(1.60)
Caries incidence (%) ^2^	17.0	23.7

^1^ Linear regression. ^2^ Logistic regression.

**Table 3 ijerph-19-16656-t003:** Adjusted (generalized linear model) model for decayed and filled surface increments of first permanent molars at the 24-months follow-up examination ( *N* = 4979 *; class = 107).

Variable	β	S.E.	*p*-Value	95%CI
Treatment				
Control group	0.27	0.09	0.001	0.11–0.44
Test group (Ref)				
Age	0.40	0.08	<0.001	0.26–0.57
Baseline DFS	0.32	0.06	<0.001	0.21–0.43
Sex				
Males	−0.44	0.08	<0.001	−0.60–0.28
Females (Ref)				
Frequency of tooth brushing				
≥2/d	−0.03	0.09	0.70	−0.21–0.14
<2/d (Ref)				
Frequency of sugar consumption				
≥1/d	0.16	0.08	0.04	0.00–0.31
<1/d (Ref)				

* In total, 5005 children completed the trial. However, there were 26 non-responses for the frequency of tooth brushing or sugar consumption.

**Table 4 ijerph-19-16656-t004:** Adjusted (generalized linear model) model for dental caries incidence of first permanent molars at the 24-months follow-up examination ( *N* = 4979 *; class = 107).

Variable	β	S.E.	*p*-Value	95%CI	OR	95%CI for OR
Treatment						
Control group	0.21	0.09	0.03	0.03–0.39	1.23	1.03–1.48
Test group (Ref)						
Age	0.35	0.08	<0.001	0.20–0.50	1.42	1.22–1.64
DFS	1.29	0.35	<0.001	0.59–1.98	3.63	1.81–7.27
Sex						
Males	−0.43	0.06	<0.001	−0.55–0.31	0.65	0.57–0.73
Females (Ref)						
Frequency of tooth brushing						
≥2/d	−0.13	0.08	0.10	−0.28–0.03	0.88	0.76–1.03
<2/d (Ref)						
Frequency of sugar consumption						
≥1/d	0.21	0.06	<0.001	0.09–0.34	1.24	1.09–1.40
<1/d (Ref)						

* In total, 5005 children participated in the trial. However, there were 26 non-responses for the frequency of tooth brushing or sugar consumption.

## Data Availability

The datasets used during the present study are available from the corresponding author upon reasonable request.

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
