# Peer review of "Effect of Fluoride Varnish in Preventing Dental Caries of First Permanent Molars: A 24-Month Cluster Randomized Controlled Trial"

_ijerph, 2022, doi:10.3390/ijerph192416656_

Round 1

Reviewer 1 Report

The study is well described. The authors explained the importance of the reduction in new caries on first molar teeth and the use of the fisrt molar teeth on the study despite another teeth.

In the discussion the authors talk about some limitations of the study, this is important. 

Author Response

We sincerely thank you for your attention to our paper. We are pleased that you have found our work interesting and valuable.

Reviewer 2 Report

Dear Editor,

Thank you for suggest me to review this paper. The article is on a very relevant topic.

I suggest that the English be revised to improve the text.

Best regards

Author Response

We sincerely thank you for your feedback and efforts to improve our work. We have improved the English language and style in the revised manuscript.

Reviewer 3 Report

I have read this paper carefully and have the following observations to make:

1.     Abstract- better contextualization of the article should be given, it is not enough to say "caries is a global health problem"

2.     Introduction- authors should describe what the literature describes the advantages and disadvantages of fluoride application.

3.     Methods- in the control group were no control consultations performed? what kind of procedure was done? How was the calibration of dentists and examiners performed?

4.     Discussion- tooth takes 5 years to complete enamel remineralization after tooth eruption. the authors must relate this factor with the results found as well as discuss other preventive treatments. Discuss the percentage of fluor and its effectiveness. Lack of adequate discussion of the limitations of the present study regarding demographic characteristics or stages of the eruption of first molars.

5.     Research should be updated or at least articles that have been recently published should be checked.

Author Response

Dear reviewer,

Thank you for your suggestion and pointed out some issues to help us improve the quality of our work. Motivated by your comments, we have deeply reconsidered the architecture and content of our work and fixed all the issues you mentioned. All changes to the manuscript were indicated in the text. We hope you will be satisfied with the revised manuscript. The following is our point-to-point responses to all of your comments:

1.Abstract- better contextualization of the article should be given, it is not enough to say "caries is a global health problem"

Response: We have provided better contextualization in the revised manuscript (Line 29-32, Page 1).

2.Introduction- authors should describe what the literature describes the advantages and disadvantages of fluoride application.

Response: We have added the literatures that describe the advantage and disadvantages of fluoride application (Line 65-67, Page2; Line 228-230, Page 5; Line 311,Page 9).

3.Methods- in the control group were no control consultations performed? what kind of procedure was done? How was the calibration of dentists and examiners performed?

Response: Oral health education was given to all children (both test group and control group) and their parents each year in class-rooms, including instructions on tooth brushing and dietary counselling. We have highlighted the information in Line 132-136, Page 3 in our manuscript.

A training and calibration exercise was undertaken in advance of each round of clinical examinations. The training session included the clinical examination of 5 children who did not participate in the study. Another 20 children were examined by the six examiners and the reference examiner with WHO criteria for calibration. Compared with the reference examiner’s results, we calculated the six examiners’ Cohen’s weighted kappa values.

  1. Discussion- tooth takes 5 years to complete enamel remineralization after tooth eruption. the authors must relate this factor with the results found as well as discuss other preventive treatments. Discuss the percentage of fluor and its effectiveness. Lack of adequate discussion of the limitations of the present study regarding demographic characteristics or stages of the eruption of first molars.

Response: We have revised the discussion section.

5.Research should be updated or at least articles that have been recently published should be checked.

Response: We have updated part of our references with more recently published literatures.

Reviewer 4 Report

This article entitled ``Effect of fluoride varnish in preventing dental caries of first 2 permanent molars: A 24-month cluster randomized controlled´´ trial aims to analyze the effectiveness of the application of topical fluoride in the prevention of caries in first permanent molars

Introduction

- Authors are recommended to carry out a more exhaustive and specific search, increasing the information presented in this section.

Material and method

- When selecting the sample, was it taken into account if the children were taking any type of medication?

If so, in what sense was it taken into account?.

- Were other factors taken into account, such as some type of disease or pathology that would make dental hygiene difficult? Were these patients excluded?

- As a result of these questions, we recommend that the authors specify in more detail the criteria for the inclusion and exclusion of the participants.

- The level of hygiene of the participants or the use of fluoride supplements at home (paste, mouthwashes, etc.) were taken into account when analyzing the results.

Results

- the authors comment that the frequency and quality of brushing and oral hygiene of the participants were analyzed. in addition to giving them instructions to improve it. Were there changes in the improvement and frequency of brushing during the study? Did the combination of improved hygiene and the presence of caries in any of the groups have any influence on the results?

- It is recommended that the authors review the data in tables 1 and 2, as well as make an epigraph with the abbreviations in all of them.

Discussion and conclusions

Within the results, the authors comment that the caries index increased in women and in patients who ate more sugars. We recommend the authors to analyze these results in this section and compare them with other studies.

As the authors comment, difficulties and possible handicaps have been encountered when carrying out this study. Will you continue to analyze and improve these aspects in the future for possible new research? Do you think the results will change once these study design issues are resolved?

Author Response

Dear reviewer:

Thank you for your suggestion and pointed out some issues to help us improve the quality of our work. Motivated by your comments, we have deeply reconsidered the architecture and content of our work and fixed all the issues you mentioned. All changes to the manuscript were indicated in the text. We hope you will be satisfied with the revised manuscript. The following is our point-to-point responses to all of your comments:

Introduction

  1. Authors are recommended to carry out a more exhaustive and specific search, increasing the information presented in this section.

Response: We have revised the introduction section according to your suggestion.

Material and method

  1. When selecting the sample, was it taken into account if the children were taking any type of medication? If so, in what sense was it taken into account?

Response: We selected children who were generally healthy and did not take any type of medication. We had provided the information in Line 117, Page3.

  1. Were other factors taken into account, such as some type of disease or pathology that would make dental hygiene difficult? Were these patients excluded?

Response: We had taken into account of these factors (Line 121, Page 3). We believe teeth with hypoplastic defects or fluorosis would be difficult to clean. Therefore, these children were excluded for participation.

  1. As a result of these questions, we recommend that the authors specify in more detail the criteria for the inclusion and exclusion of the participants.

Response: We have revised the criteria to make them more clear to read (Line 117-122, Page 3).

  1. The level of hygiene of the participants or the use of fluoride supplements at home (paste, mouthwashes, etc.) were taken into account when analyzing the results.

Response: All children were encouraged to brush their teeth twice a day with fluoride toothpaste throughout the study (Line 134, Page 3). However, whether they strictly followed the requirements or whether the child receive any other dental preventive interventions or treatment in dental clinics were not followed or recorded. Therefore, it was unable to assess how these treatments would influence the outcome of the study. We had provided the limitation in the discussion section (Line 296-299, Page 8).

Results

  1. The authors comment that the frequency and quality of brushing and oral hygiene of the participants were analyzed. in addition to giving them instructions to improve it. Were there changes in the improvement and frequency of brushing during the study? Did the combination of improved hygiene and the presence of caries in any of the groups have any influence on the results?

Response: Questionnaire was only required and analyzed at baseline in our study design. Therefore, we could hardly assess the change of child’s oral health-related behavior (frequency of tooth brushing and sweets intake) after 24-month intervention and their influence on the study results. This is one of the limitations of our trial. We have added this limitation in the discussion section (Line 298, Page 8).

  1. It is recommended that the authors review the data in tables 1 and 2, as well as make an epigraph with the abbreviations in all of them.

Response: We have reviewed the data in table 1 and 2. The epigraph of the abbreviation had been given in Line 181, Page 4.

Discussion and conclusions

  1. Within the results, the authors comment that the caries index increased in women and in patients who ate more sugars. We recommend the authors to analyze these results in this section and compare them with other studies.

Response: The aim of the study was to evaluate the caries-preventive effect of semi-annual application of fluoride varnish on first permanent molars. The relationship between these factors (age, sex, baseline caries experience, frequency of tooth brushing, and frequency of sugar consumption are factors that may related to dental health) and dental caries have been discussed over lots of epidemiological studies. We believe it was not the point we should focus on in this study. Therefore, we did not discuss them.

  1. As the authors comment, difficulties and possible handicaps have been encountered when carrying out this study. Will you continue to analyze and improve these aspects in the future for possible new research? Do you think the results will change once these study design issues are resolved?

Response: Of course. We will continue our study on caries prevention all the way. We will improve our study design in future to avoid the limitations in this trial. We believe our study may have already provided some evidence to support some reasonable assumptions. Once the study design improved, the data of prevention fraction may be fluctuated (Line329-332, Page 9).

Round 2

Reviewer 4 Report

Thank you very much for your answers. Congratulations for the work done

Author Response

We sincerely thank you and your valuable feedback. We are pleased to learn that you have found our research work interesting and also pointed out some issues to help us improve the quality of our work.  We have carefully checked all our references again, we believe they are all relevant to the research.